# Anti-Tumor Effect of Inhibition of DNA Damage Response Proteins, ATM and ATR, in Endometrial Cancer Cells

**DOI:** 10.3390/cancers11121913

**Published:** 2019-12-01

**Authors:** Makoto Takeuchi, Michihiro Tanikawa, Kazunori Nagasaka, Katsutoshi Oda, Yoshiko Kawata, Shinya Oki, Chuwa Agapiti, Kenbun Sone, Yuko Miyagawa, Haruko Hiraike, Osamu Wada-Hiraike, Hiroyuki Kuramoto, Takuya Ayabe, Yutaka Osuga, Tomoyuki Fujii

**Affiliations:** 1Department of Obstetrics and Gynecology, Graduate School of Medicine, The University of Tokyo, Tokyo 113-8655, Japan; matakeuchi-tky@umin.ac.jp (M.T.); tanikawa-tky@umin.ac.jp (M.T.); katsutoshi-tky@umin.ac.jp (K.O.); yoshiko_kawata@hotmail.com (Y.K.); rectus@world.odn.ne.jp (S.O.); agapitichuwa@yahoo.com (C.A.); sonekenbun@hotmail.co.jp (K.S.); osamuwh-tky@umin.ac.jp (O.W.-H.); yutakaos-tky@umin.ac.jp (Y.O.); fujiit-tky@umin.ac.jp (T.F.); 2Department of Obstetrics and Gynecology, Teikyo University School of Medicine, Tokyo 173-8605, Japan; m.yuko0201@gmail.com (Y.M.); hharu-tky@umin.ac.jp (H.H.); tayabe@med.teikyo-u.ac.jp (T.A.); 3Kanagawa Health Service Association, Yokohama 231-0021, Japan; kuramoto@yobouigaku-kanagawa.or.jp

**Keywords:** DNA damage response (DDR) pathways, ATR-Chk1 pathway, ATM-Chk2 pathway, combination therapy, molecular-targeted therapies

## Abstract

While the incidence of endometrial cancer continues to rise, the therapeutic options remain limited for advanced or recurrent cases, and most cases are resistant to therapy. The anti-tumor effect of many chemotherapeutic drugs and radiotherapy depends on the induction of DNA damage in cancer cells; thus, activation of DNA damage response (DDR) pathways is considered an important factor affecting resistance to therapy. When some DDR pathways are inactivated, inhibition of other DDR pathways can induce cancer-specific synthetic lethality. Therefore, DDR pathways are considered as promising candidates for molecular-targeted therapy for cancer. The crosstalking ataxia telangiectasia mutated and Rad3 related and checkpoint kinase 1 (ATR-Chk1) and ataxia telangiectasia mutated and Rad3 related and checkpoint kinase 2 (ATM-Chk2) pathways are the main pathways of DNA damage response. In this study, we investigated the anti-tumor effect of inhibitors of these pathways in vitro by assessing the effect of the combination of ATM or ATR inhibitors and conventional DNA-damaging therapy (doxorubicin (DXR), cisplatin (CDDP), and irradiation) on endometrial cancer cells. Both the inhibitors enhanced the sensitivity of cells to DXR, CDDP, and irradiation. Moreover, the combination of ATR and Chk1 inhibitors induced DNA damage in endometrial cancer cells and inhibited cell proliferation synergistically. Therefore, these molecular therapies targeting DNA damage response pathways are promising new treatment strategies for endometrial cancer.

## 1. Introduction

Endometrial cancer represents the most common gynecologic malignancy in developed countries with approximately 170,000 cases diagnosed yearly, resulting in almost 36,000 deaths annually [1]. Endometrial cancer is classified into two pathological types, type 1 (estrogen-dependent) and type 2 (estrogen-independent). Endometrioid carcinoma (grades 1 and 2) is classified as type 1 and accounts for 80% of the endometrial cancer cases. Poorly differentiated endometrioid carcinoma, serous carcinoma, clear-cell carcinoma, and undifferentiated carcinoma are less frequent, and classified as type 2 [2]. The carcinogenesis of type 1 endometrial cancer is related to a continuous exposure to estrogen, e.g., early menarche, late menopause, nulliparous, hormone replacement therapy, obesity, hypertension, and diabetes [3,4,5]. Although the peak age of onset range is 50–60 years, the incidence of endometrial cancer has been rising in all age groups [6]. The standard treatment of endometrial cancer consists of surgery followed by radiation therapy, chemotherapy, or both, depending on the risk factors of a patient, which include muscle layer invasion, vascular invasion, grade, lymph node metastasis, and the stage of the disease [7,8,9,10]. Doxorubicin (DXR) and cisplatin (CDDP) are the most commonly used chemotherapeutics. Nonetheless, 10–15% of the cases without adjuvant therapy have a recurrence of cancer. The prognosis for recurrent cases is very poor, with a 5-year survival rate of approximately 15% because of the limited choices of treatment and resistance to therapy. Therefore, molecular-targeted therapy is expected to be the new strategy for endometrial cancer treatment [11,12].

DNA damage response (DDR) proteins are activated or inactivated in various types of cancer [13]. The anti-tumor effect of many chemotherapeutic drugs and radiotherapy depends on the induction of DNA damage to cancer cells, for example, DXR increases double-strand breaks through inhibition of topoisomerase II, CDDP causes DNA crosslinks and nucleotide excision repair, and irradiation induces DNA damage directly. The activation of DDR proteins in cancer cells is regarded as one of the most important factors of resistance to therapy [14]. On the other hand, in case some of the DDR pathways are inactivated, cancer cells may depend more highly on another DDR pathway, and inhibiting the other DDR pathway can induce cancer-specific synthetic lethality [15,16,17]. Therefore, DDR pathways are considered as promising candidates for molecular-targeted therapy.

Additionally, endometrial cancer cells can be in a state of genomic instability with replication stress through loss of *Phosphatase and tensin homolog* (*PTEN*) or *Ras* mutations. Endometrial cancer cells may survive such instability by activating DDR pathways, therefore molecular therapy targeting the DDR pathway may be highly effective [18]. ATR (ATM and Rad3-related) and ATM (ataxia telangiectasia-mutated) proteins are members of the PIKK (PI3K-like protein kinase) family, which are autophosphorylated and activated by DNA damage; then, they induce phosphorylation of its downstream target, such as Chk1 (checkpoint kinase 1) or Chk2 (checkpoint kinase 2), and regulate cell cycle and DNA repair. *ATR* is known as the gene responsible for Seckel syndrome, a rare disorder that typically results in short stature (dwarfism) [19]. ATR mainly reacts to ultraviolet light or a single-strand DNA generated by a DNA replication disorder, and subsequently undergoes autophosphorylation, induces phosphorylation of its downstream target Chk1, and regulates cell cycle [20,21]. Meanwhile, ATM was identified as the responsible gene of ataxia telangiectasia, a hereditary disease characterized by cancer predisposition [22]. ATM mainly reacts to DNA double-strand breaks and, like ATR, it then induces autophosphorylation and phosphorylation of Chk2 and p53, and regulates cell cycle [20,21]. ATR-Chk1 and ATM-Chk2 pathways mutually crosstalk [23]. Clinical trials are now ongoing to evaluate the use of inhibitors of these pathways in some types of cancers [24]. Regarding the effect of ATR inhibitors and ATM inhibitors in endometrial cancer, the ATR inhibitor ETP46464 enhances the anti-tumor effect of CDDP in endometrial cancer cells, but the ATM inhibitor KU55933 does not, while both the inhibitors enhance the effect of irradiation [25]. There is no previous report about the effect of inhibitors of these pathways combined with DXR. The ATM-Chk2 pathway frequently has mutations or decreased expression, not only in hereditary cases, but also in many types of cancers, such as endometrial cancer, breast cancer, pancreas cancer, head and neck squamous cell cancer, and non-small-cell lung cancer. Thus, inhibiting the ATR-Chk1 pathway may induce cancer-specific synthetic lethality [26,27,28,29,30,31,32]. Recently, it was reported that the combination of an ATR inhibitor and a Chk1 inhibitor induces cancer-specific cell death in breast cancer cells and osteosarcoma cells [33], but there is no report about its effect on endometrial cancer cells. Therefore, in this study, we aimed to clarify the anti-tumor effect of an ATR inhibitor or an ATM inhibitor combined with DXR, CDDP, or irradiation, and if the combination of the ATR inhibitor and the Chk1 inhibitor could induce DNA damage in endometrial cancer cells.

## 2. Results

### 2.1. The Effect of ATR Inhibitor or ATM Inhibitor Combined with DXR or CDDP in Endometrial Cancer Cells

We performed a cell viability assay to determine the half maximal inhibitory concentration (IC_50_) value of VE822 (ATR inhibitor) and KU60019 (ATM inhibitor). Figure 1 shows the cell viability in various concentrations of the inhibitors. The IC_50_ of VE822 was 1.5 µM and that of KU60019 was 20 µM.

Next, we performed immunoblotting to verify if the ATR or ATM pathway is activated by exposure to DXR or CDDP. The phosphorylation of Chk1 on Ser345 was considered as the index of ATR activation [34]. The phosphorylation of ATM on Ser1981 and that of Chk2 on Thr68 were considered as indices of ATM activation [35,36]. γH2AX was considered as the index of DNA double-strand breaks, and β-actin was the control. The exposure to both DXR and CDDP increased the expression of p-Chk1, p-ATM, p-Chk2, and p-H2AX. Moreover, the addition of the ATR inhibitor (10–1000 nM) decreased the expression of p-Chk1 (Figure 2a) and the addition of the ATM inhibitor (1–100 µM) decreased the expression of p-ATM and p-Chk2 (Figure 2b).

In light of these results, we then performed a clonogenic assay to examine the anti-tumor effect of the combination of a conventional chemotherapeutical drug and the ATR or ATM inhibitor. The combination of DXR (10–50 μM) or CDDP (100–500 nM) (Figure 3a) and VE822 (10–50 nM) or KU60019 (100–1000 nM) showed a remarkable enhancement of cell killing (Figure 3b).

Furthermore, we performed flow cytometry to examine the cell cycle distribution after treatment with these chemotherapeutic drugs and inhibitors. As shown in Figure 4a, the combination of DXR and KU60019 remarkably increased the sub-G1 population to 24.7% compared to DXR alone (10.1%) or KU60019 alone (6.0%) on HEC-1B cells, and the combination of DXR and KU60019 increased the sub-G1 population to 14.0% compared to DXR alone (4.8%) or KU60019 (9.1%) alone on HEC-6 cells. When combining CDDP and VE822, there was a marked increase in the sub-G1 population to 29.8% compared to CDDP alone (4.0%) or VE822 alone (5.9%) on HEC-1B cells. On HEC-6 cells, interestingly, this combination increased the sub-G1 population to 20.3% compared to CDDP alone (4.5%) or VE822 alone (4.9%) (Figure 4b).

### 2.2. The Effect of ATR Inhibitor or ATM Inhibitor Combined with Irradiation in Endometrial Cancer Cells

We performed immunoblotting to examine the activation caused by irradiation and the inactivation with inhibitors of the ATR-Chk1 and ATM-Chk2 pathways. To examine the change over time in the activation of the pathway, we extracted proteins from cells that had been irradiated for periods ranging from 15 min to 72 h. Both the pathways were activated 2 h after irradiation (Figure 5a). Irradiation increased the expression of p-Chk1, p-ATM, p-Chk2, and p-H2AX. Moreover, adding the ATR inhibitor VE822 (1 µM) decreased the expression of p-Chk1 (Figure 5b) and adding the ATM inhibitor KU60019 (10 µM) decreased the expression of p-ATM, p-Chk2, and p-H2AX (Figure 5c). 

Next, we performed a clonogenic assay to examine the anti-tumor effect of the combination of irradiation and the inhibitors. Both VE822 (50 nM) and KU60019 (1 µM) enhanced the effect of irradiation (2–8 Gy) (Figure 6).

### 2.3. The Effect of the Combination of ATR Inhibitor and Chk1 Inhibitor in Endometrial Cancer Cells

Additionally, we performed a cell viability assay to examine the effect of the ATR inhibitor and a Chk1 inhibitor in endometrial cancer cells. Figure 7 shows the cell viability after exposure to different concentrations of the inhibitors. The IC_50_ of VE822 was 1.5 µM and that of KU60019 was 20 µM. We used AZD7762 as the Chk1 inhibitor, and evaluated the synergy by calculating the combination index by the Talalay-Chou method [37]. The two inhibitors are synergistically effective if the combination index is 1 or less. The combination index of VE822 and AZD7763 (50 nM) was 0.43 in HEC-1B cells, and 0.28 with HEC-6 cells, thus these inhibitors had a synergistic effect in both the cell lines (Figure 7).

Next, we performed immunoblotting and immunofluorescence to examine the DNA damage caused by the ATR and the CHk1 inhibitors. The combination of VE822 (1 µM) and AZD7762 (30–60 nM) increased the expression of p-H2AX (Figure 8a). The results of the immunofluorescence experiment revealed that the combination of VE822 (1 µM) and AZD7762 (60 nM) induced nuclear accumulation of p-H2AX and BRCA1 (Figure 8b).

## 3. Discussion

In this study, we examined the anti-tumor effect of the inhibitors of the ATR-Chk1 or ATM-Chk2 pathway in endometrial cancer cells treated with DXR or CDDP, used as standard chemotherapeutic drugs in endometrial cancer treatment. As exposure to both DXR and CDDP increased the expression of p-H2AX in immunoblotting, it is assumed that these drugs induced DNA damage to endometrial cancer cells. Additionally, as both DXR and CDDP increased the expression of p-Chk1, p-ATM, and p-Chk2, it is possible that both ATR-Chk1 and ATM-Chk2 pathways were activated in response to the DNA damage.

The ATR inhibitor, VE822, decreased the expression of p-Chk1, which confirmed the inhibition of ATR activation. Similarly, the ATM inhibitor, KU60019, decreased the expression of p-ATM and p-Chk2, confirming that ATM activation was inhibited. The expression of p-H2AX was decreased in the presence of KU60019, because H2AX was phosphorylated by activated ATM. VE822 (1000 nM) and KU60019 (100 µM) increased p-H2AX expression, because cells were injured by each single agent at these high concentrations. The combination of DXR and VE822 decreased p-ATM and p-CHk2 and the combination of DXR and KU60019 decreased p-Chk1. On the other hand, the combination of CDDP and VE822 did not decrease p-ATM or p-CHk2, and the combination of CDDP and KU60019 did not decrease p-CHk1. These differences may be due to the difference of the mechanism of DNA damage induced by DXR or CDDP, or due to the crosstalk between the ATR-Chk1 and ATM-CHk2 pathways.

The IC_50_ of VE822 was 1.5 µM and that of KU60019 was 20 µM, and these results demonstrated that VE822 inhibited the ATR-Chk1 pathway and KU60019 inhibited the ATM-Chk2 pathway at a concentration that did not affect cell viability. In the clonogenic assay, every combination of DXR or CDDP and VE822 (50 nM) or KU60019 (1 μM) showed a synergistic effect. The sub-G1 population of the combination group was increased in the cell cycle assay, suggesting that the synergy resulted from the induction of cell death.

Considering the results discussed above, we proved that the ATR inhibitor, VE822, inhibited the activation of the ATR-Chk1 pathway induced by DXR or CDDP and that the ATM inhibitor, KU60019, inhibited the activation of the ATM-Chk2 pathway induced by DXR or CDDP. Moreover, both the inhibitors enhanced the effect of DXR or CDDP in endometrial cancer cells. Further, we examined the effect of the combination of irradiation and these inhibitors. In immunoblotting, p-H2AX was increased in the irradiated group; therefore, DNA damage was considered to be induced in endometrial cancer cells; p-Chk1, p-ATM, and p-Chk2 were increased in the irradiated group, which indicated that the ATR-Chk1 and ATM-Chk2 pathways were activated in response to DNA damage. Additionally, p-Chk1 was decreased after the addition of VE822, and p-ATM and p-Chk2 followed the same trend after the addition of KU60019; therefore, we concluded that the ATR and ATM drugs inhibited the ATR-Chk1 and ATM-Chk2 pathways, respectively, which were activated by irradiation. In addition, the decrease in p-H2AX expression was due to inhibition of the activation of ATM by KU60019. Although VE822 did not decrease p-ATM or p-Chk2, KU60019 decreased p-Chk1. This may be the result of the crosstalk between the two pathways. In the clonogenic assay, the combination of irradiation and VE822 (50 nM) or KU60019 (1 μM) had a synergistic effect on cell killing. For the reason above, we concluded that both the inhibitors enhanced the effect of irradiation in endometrial cancer cells: the ATR inhibitor, VE822, inhibited the activation of the ATR-Chk1 pathway induced by irradiation; and the ATM inhibitor, KU60019, inhibited the activation of the ATM-Chk2 pathway, also caused by irradiation.

It is well known that various cancers harbor somatic mutations in the ATM-Chk2-p53 pathway. Thus, inhibiting the ATR-Chk1 pathway rather than the ATM-Chk2-p53 pathway may induce cancer-specific cell death, such as synthetic lethality. In this study, we examined the effect of the combination of the ATR inhibitor, VE822, and the Chk1 inhibitor, AZD7762. The results of the cell viability assay showed that there was a synergistic anti-tumor effect in endometrial cancer cells. Moreover, the immunoblotting analysis revealed that p-H2AX was increased, and in the immunofluorescence assay, p-H2AX and BRCA1 were found to be accumulated in the cell nuclei in the dual inhibition group. In endometrial cancer, Chk1 prevents unscheduled origin firing in concert with suppressing CDK, which is activated by oncogenic stress in a manner similar to that occurring in other cancer cells [33]. Thus, Chk1 inhibition induces excessive origin firing and elevates replication stress to increase single-strand DNA (ssDNA) in the stalled replication forks. In contrast, ATR inhibition facilitates the inhibition of replication protein A (RPA) supply, resulting in the collision of stalled replication forks and DNA double-strand breaks. Therefore, we believed that the combination of the ATR inhibitor and the Chk1 inhibitor induced DNA damage in endometrial cancer cells. As for the effect of the ATR inhibitor or the ATM inhibitor in endometrial cancer, a previous report showed that the ATR inhibitor ETP46464 enhanced the effect of CDDP in endometrial cancer cells, but the ATM inhibitor KU55933 did not [27]. While ETP46464 inhibited not only ATR, but also mammalian target of rapamycin (mTOR), DNA-dependent protein kinase (DNA-PK), phosphoinositide 3-kinases alpha (PI3Kα), and ATM [38,39,40], VE822 used in this study had higher selectivity to ATR, showing effects even at low concentrations. Similarly, KU60019 is an improved analog of KU55933 with demonstrated high selectivity to ATM and pronounced effect at low concentrations [41,42,43]. Thus, the enhancement of the anti-tumor effect of CDDP and DXR by VE822 and KU60019 was probably due to the high selectivity of these improved inhibitors. Therefore, adding these inhibitors to conventional chemotherapy can be a promising therapeutic option for endometrial cancer. It was reported previously that both ETP46464 and KU55933 enhanced the effect of irradiation, and that the combination of these two inhibitors more strongly enhanced this effect [25]. We did not examine the effect of the combination of the ATR inhibitor and the ATM inhibitor, but adding both VE822 and KU60019 may more strongly enhance the anti-tumor effect of irradiation.

In a previous publication, ATR mutation, often associated with poor prognosis, was identified in approximately 5% of the endometrial cancer cases. Moreover, ATR mutation was identified significantly in microsatellite instability-positive cases and as an independent factor of poor prognosis [44]. In these cases, some pathway other than ATR may compensate the DNA damage response. On the other hand, ATM expression was significantly decreased as the process of the carcinogenesis advanced in the endometrium [45]. Thus, a functional decline of ATM and ATR was associated with the process of the carcinogenesis or malignancy of endometrial cancer. Considering the biological feature of endometrial cancer in terms of failure of the DNA response pathway associated with carcinogenesis, DNA-damaging conventional therapy combined with molecular-targeted drugs of the DNA response pathway may induce new synthetic lethality.

The main limitation of our study was that we did investigate these effects in vivo. A previous publication demonstrated that VE822 combined with gemcitabine or irradiation suppressed tumor growth in a pancreatic cancer xenograft animal model, where the rate of oral administration was 60 mg/kg [39]. In addition, KU60019 combined with irradiation prolonged the overall survival of mice with glioma xenografts that received 10 µM (osmotic pumps) or 250 µM (conventional enhanced delivery) of the drug [43]. No adverse event was reported in both the studies. In our in vitro study, the concentrations which enhanced the effect of DXR, CDDP, or irradiation in the clonogenic assay were 10–50 nM (VE822) and 100–1000 nM (KU60019). These concentrations were significantly lower than those used in these previous reports, thus considered as feasible for safe administration in an in vivo setting. Further examination using endometrial cancer xenografts in vivo to evaluate the effectiveness or adverse effect is warranted. 

The reason for the enhancement of the sensitivity to these DNA-damaging agents or irradiation is probably related to the DNA damage that remained unrepaired with these inhibitors; subsequently, cells deviated from the cell cycle checkpoints, resulting in cell death. HEC-1B cells have mutant p53 and HEC6 cells have wild-type p53. Considering that both ATR and ATM inhibitors enhanced the effect of the DNA-damaging agents and irradiation in both the cell lines, p53 pathway and other pathways seem to work cooperatively. Not only the status of p53, but also that of p53-independent pathways can be biomarkers of these inhibitors. More studies are needed to elucidate how the ATR-Chk1 pathway and the ATM-Chk2 pathway are used separately, and the type of crosstalk occurring between the two pathways. Moreover, additional studies are expected to reveal the most appropriate therapy to combine with these inhibitors in clinical applications (such as irradiation, DNA-damaging agents, or other molecular-targeted drugs) as well as the most suitable biomarker for such treatments.

## 4. Materials and Methods

### 4.1. Cell Culture and Reagents

The HEC-1B uterine endometrial cancer cell line was obtained from the American Type Culture Collection (Manassas, VA, USA) and HEC-6 cell line was established by Hiroyuki Kuramoto (Kanagawa Health Service Association, Japan). Both cell lines were cultured in Eagle’s minimal essential medium supplemented with 10% fetal bovine serum in a humidified incubator at 37 °C and 5% CO_2_. KU60019 (ATM inhibitor) and VE822 (ATR inhibitor) were purchased from Selleck (Houston, TX, USA). Both inhibitors were dissolved in DMSO (10 mM stock solutions) and stored at −80 °C.

### 4.2. Cell Viability Assay

Cells (2 × 10^4^ cells/mL, 100 μL/well) were seeded in 96-well plates with medium and allowed to attach for 24 h. The medium was replaced with fresh medium containing KU60019 (0–10 μM) or VE822 (0–10 μM) for 1 h before doxorubicin (0–2.5 μM) or cisplatin (0–10 μM) treatment. Seventy-two hours later, 10 µL of Cell Count Kit-8 solution (Dojindo, Tokyo, Japan) was added to each well for 2 h for the methyl thiazolyl tetrazolium (MTT) assay. The change in absorbance at 450 nm was measured on a microplate reader (BioTek, Winooski, VT, USA); cells treated with DMSO were used for normalization. The experiment was repeated three times.

### 4.3. Clonogenic Assay

Cells were seeded in 6-well plates at a density of 1000 cells/well and allowed to attach for 24 h. Then, cells were incubated with KU60019 (0–1 μM) or VE822 (0–50 nM) for 1 h before adding doxorubicin (0–50 nM) or cisplatin (0–500 nM) or X-ray exposure (0–8 Gy) with Pantac HF350 (Shimadzu, Kyoto, Japan). After additional 7–10 days, i.e., the time required for a mean colony size of ≥50 cells, cells were fixed with methanol and stained with Giemsa. Colonies with ≥50 cells were counted manually. The cell survival percentage was calculated as the surviving fraction of cells in the presence of inhibitors for any given dose or concentration of the damaging agent divided by the surviving fraction of cells treated with DMSO alone.

### 4.4. Immunoblotting

Equal amounts of proteins were fractionated by SDS-PAGE and transferred onto a polyvinylidene difluoride membrane (Millipore, Bedford, MA, USA). The membranes were blocked and primary antibodies were added, followed by secondary antibodies. Signals were detected using an ImageQuant LAS 4000 Mini instrument (GE Healthcare, Wauwatosa, WI, USA). 

HRP (horseradish peroxidase)-linked antibodies specific for ATR, phospho-ATR (Ser428), phosphor-Chk1 (Ser345), ATM, phospho-Chk2 (Thr68), phospho-H2AX (Ser139), and anti-mouse or anti-rabbit IgG were obtained from Cell Signaling Technology (Beverly, MA, USA). Antibodies specific for Chk1 were purchased from Santa Cruz Biotechnology (Santa Cruz, CA, USA), pATM from GeneTex (Irvine, CA, USA), Chk2 from Medical and Biological Laboratories (Nagoya, Japan), and β-actin from Sigma-Aldrich (St. Louis, MO, USA). Antibodies were used according to the manufacturers’ recommendations. 

Protein bands were detected by the ECL Western Blot Detection Kit (GE Healthcare Life Sciences, Piscataway, NJ, USA).

Molecular weight markers were used and all gel images with the markers are shown in the Appendix A. 

### 4.5. Cell Cycle Analysis

Cells (2 × 10^5^ cells/well) were seeded on 60 mm dishes. After 24 h of incubation, the medium was replaced with fresh medium containing various concentrations of the drugs for 72 h. Cells were collected by using trypsin and stained with propidium iodide (PI; 50 µg/mL) (Sigma-Aldrich) at 4 ℃ for 30 min in the dark. Cell cycle distribution was analyzed by flow cytometry on an Epics XL instrument (Beckman Coulter, Brea, CA, USA) using the Cell Quest Pro software v3.1. (BD Bioscience, Franklin Lakes, NJ, USA) according to the manufacturers’ recommendations. The experiment was repeated three times.

### 4.6. Statistical Analysis

The significance of the differences between the means was measured by one-tailed *t*-test. The values were expressed as means ± the standard error of the mean (SEM). A *p*-value < 0.05 was considered as statistically significant.

## 5. Conclusions

In this study, we investigated the anti-tumor effect of inhibitors of the ATR-Chk1 and ATM-Chk2 pathways in endometrial cancer cells. In particular, we evaluated the effect of the combination of an ATM or ATR inhibitor and conventional DNA-damaging therapy (doxorubicin, cisplatin, or irradiation) and found that both the inhibitors enhanced the cell sensitivity to the three conventional therapies. Moreover, the combination of the ATR and Chk1 inhibitors induced DNA damage in endometrial cancer cells and inhibited cell proliferation synergistically. Our results showed that DNA damage response pathway-targeting molecular therapies are promising treatment strategies for endometrial cancer.

## Figures and Tables

**Figure 1 cancers-11-01913-f001:**
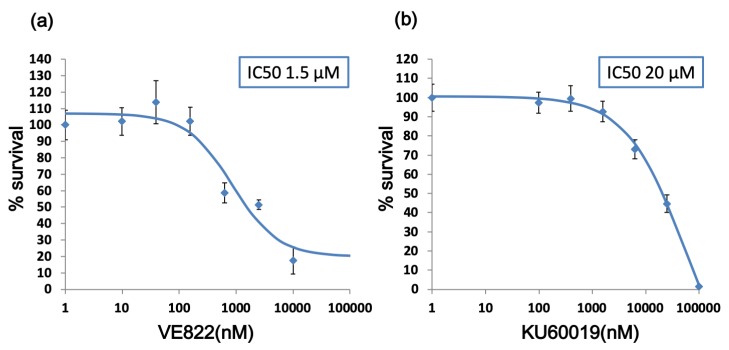
Toxicity of KU60019 or VE822 to endometrial cancer cells. Cell viability assay showing the sensitivity of endometrial cancer cells, HEC-6, to VE822 (**a**) or KU60019 (**b**) drug at increasing concentrations for 72 h. The data were normalized to the value of control cells. Each value is shown as the mean of three experiments ±SD.

**Figure 2 cancers-11-01913-f002:**
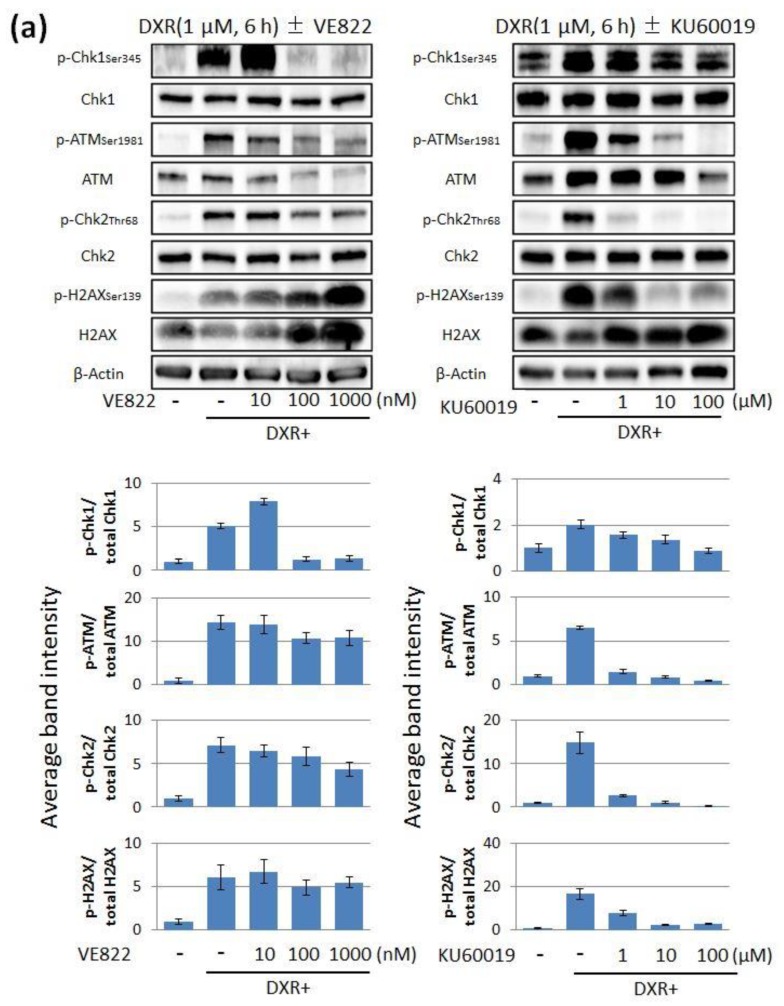
DXR and CDDP activated both ATR/Chk1 and ATM/Chk2 pathways in HEC-6 endometrial cancer cells, which was canceled by each inhibitor. The medium was replaced with fresh medium containing (**a**) VE822 (0–1000 nM) or (**b**) KU60019 (0–100 μM) for 1 h before DXR (1 μM, 6 h) or CDDP (20 μM, 6 h) treatments. DXR, doxorubicin; CDDP, cisplatin. The lower histograms show the quantitative analyses of the intensities of the phosphoprotein bands from three independent experiments with SD indicated. A Western blot showing all the bands with all the molecular weight markers is in Appendix A.

**Figure 3 cancers-11-01913-f003:**
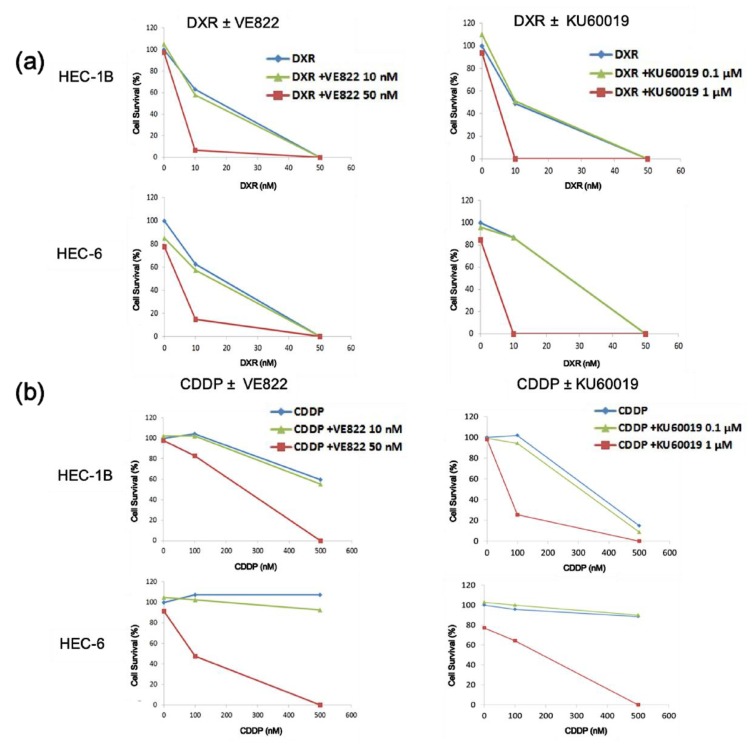
Both ATR and ATM inhibitors sensitize endometrial cancer cells to cell death. HEC-1B or HEC-6 cells were exposed to (**a**) DXR (10–50 μM) or (**b**) CDDP (100–500 nM) after incubation with VE822 (10–50 nM) or KU60019 (100–1000 nM) for 1 h, and cultured for 7–10 days. Colonies with ≥50 cells were counted and the cell survival percentage was calculated. Cells treated with dimethyl sulfoxide (DMSO) were used for normalization. DXR, doxorubicin; CDDP, cisplatin.

**Figure 4 cancers-11-01913-f004:**
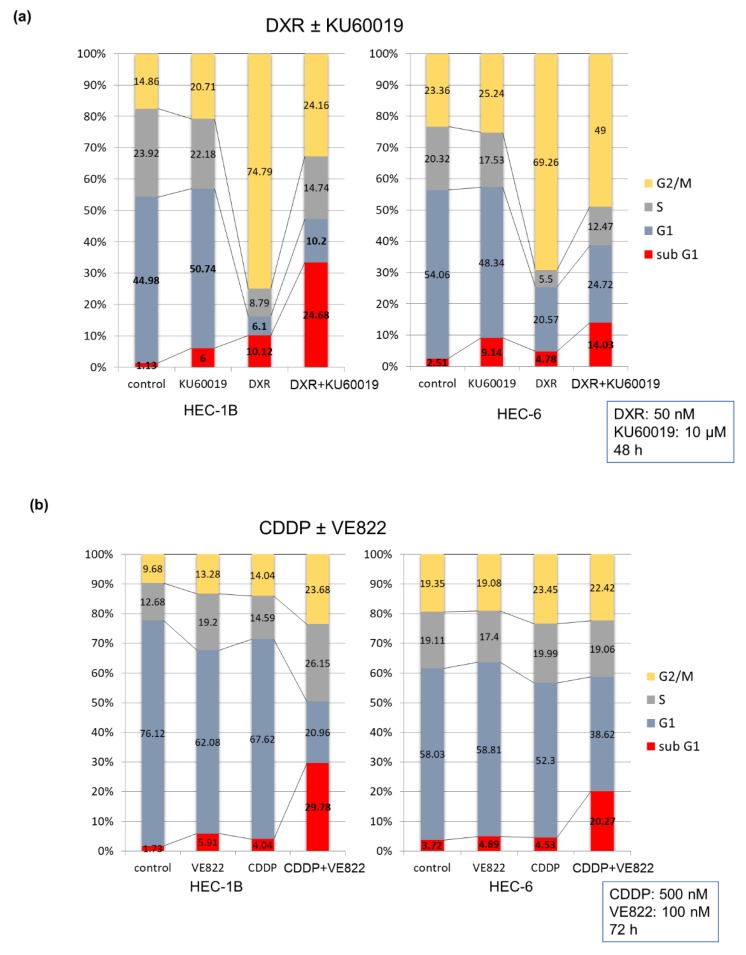
Flow cytometric analysis of cell cycle in endometrial cancer cells treated with the combination of DNA-damaging agents and inhibitors. Cells were treated with the drugs for 48 or 72 h and cell cycle distribution was analyzed by flow cytometry. (**a**) DXR (50 nM) and KU60019 (10 μM) incubated for 48 h. (**b**) CDDP (500 nM) and VE822 (100 nM) incubated for 72 h. DXR, doxorubicin; CDDP, cisplatin. The raw data is in Appendix A.

**Figure 5 cancers-11-01913-f005:**
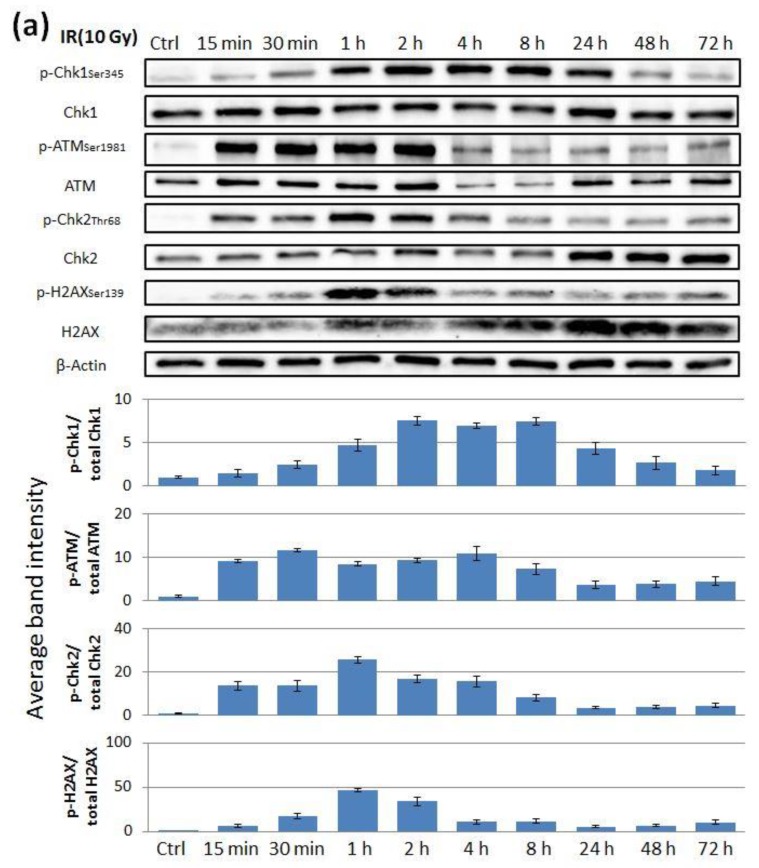
Irradiation activated both ATR/Chk1 and ATM/Chk2 pathways in HEC-6 endometrial cancer cells, which was then canceled by each inhibitor. (**a**) Proteins were extracted from HEC-6 cells after irradiation for a period from 15 min to 72 h. The medium was replaced by fresh medium, and the inhibitor (**b**) VE822 (0–1000 nM) or (**c**) KU60019 (0–10 μM) was added for 1 h before irradiation (10 Gy). All proteins were extracted 2 h after irradiation. The lower histograms show the quantitative analyses of the intensities of the phosphoproteins bands from three independent experiments with SD indicated. A Western blot showing all the bands with all the molecular weight markers is in Appendix A.

**Figure 6 cancers-11-01913-f006:**
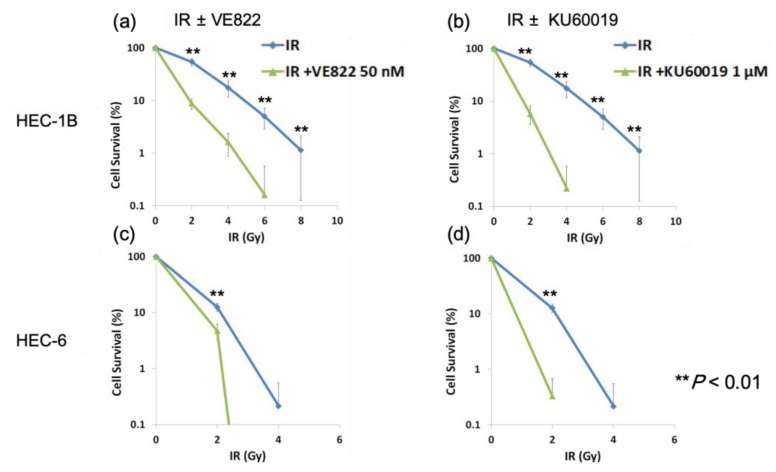
ATR inhibitor or ATM inhibitor sensitizes endometrial cancer cells to irradiation. HEC-1B or HEC-6 cells were exposed to IR (2–8 Gy) after incubation with VE822 (50 nM) or KU60019 (1 μM) for 1 h, and cultured for 7–10 days. (**a**) HEC-1B (IR after incubation with VE822). (**b**) HEC-1B (IR after incubation with KU60019). (**c**) HEC-6 (IR after incubation with VE822). (**d**) HEC-6 (IR after incubation with KU60019). Colonies with ≥50 cells were counted and the cell survival percentage was calculated. Cells treated with DMSO were used for normalization. IR, irradiation.

**Figure 7 cancers-11-01913-f007:**
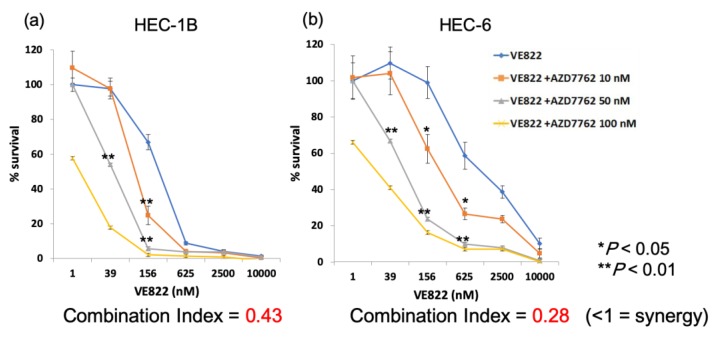
Cell viability assay to evaluate the combination of an ATR inhibitor and a Chk1 inhibitor. HEC-1B or HEC-6 cells were exposed to VE822 (39–10000 nM) and AZD7762 (10–100 nM) for 72 h before measuring cell viability. (**a**) HEC-1B with VE822 and AZD7762. (**b**) HEC-6 with VE822 and AZD7762. The data were normalized to the value of control cells. Each value is shown as the mean of three experiments ±SD.

**Figure 8 cancers-11-01913-f008:**
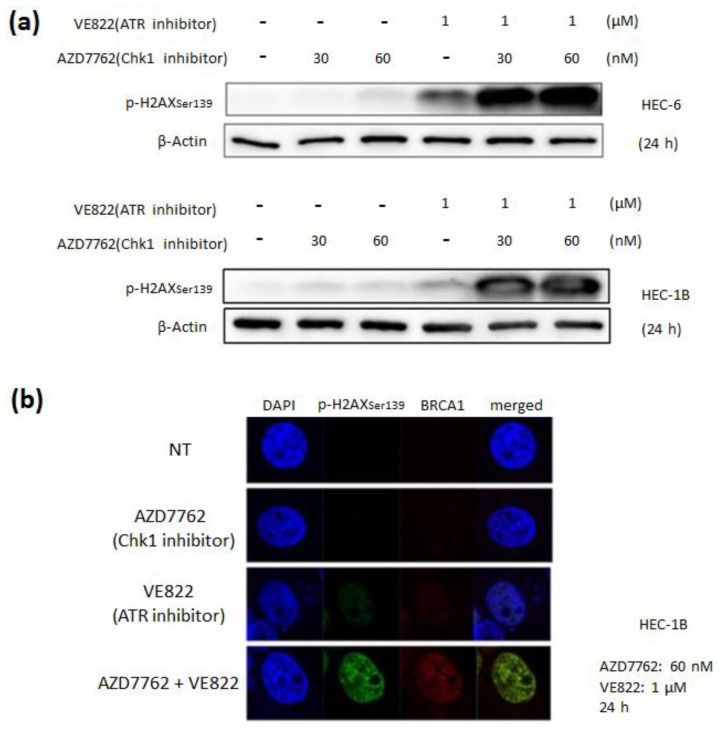
Evaluation of the effect of the combination of the ATR inhibitor and the Chk1 inhibitor by immunoblotting and immunofluorescence. (**a**) HEC-6 (upper figure) and HEC-1B (lower figure) were treated with VE822 (1 μM) and AZD7762 (30–60 nM) for 24 h before protein extraction. A Western blot showing all the bands with all the molecular weight markers is in Appendix A. (**b**) HEC-1B cells were treated with VE822 (1 μM) and AZD7762 (60 nM) for 24 h before cell fixation and immunofluorescence analysis.

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
