# Peer review of "Anti-Tumor Effect of Inhibition of DNA Damage Response Proteins, ATM and ATR, in Endometrial Cancer Cells"

_cancers, 2019, doi:10.3390/cancers11121913_

Round 1
Reviewer 1 Report
"Anti-tumor effect of inhibition of DNA damage 3 response proteins, ATM and ATR, in endometrial 4 cancer cells" manuscript needs to be improved methodologically. Major concerns are:
in figures 2 and 5 ß actin shows that leading was not optimized Figure 4 is not the way to present flow cytometry data, please show "raw FACS data" some curves in figures show statistical significance, some don't HEC1B are mentioned only in methods, otherwise in results are refered only as endometrial cancer cells, which is not very informative
Author Response
November 25th, 2019
Prof. Dr. Samuel C. Mok
Editor-chief
Cancers
We would like to resubmit our manuscript entitled “Anti-tumor effect of inhibition of DNA damage response proteins, ATM and ATR, in endometrial cancer cells” (Manuscript ID: cancers-647276). We thank the editor and the reviewers for carefully reviewing our manuscript and for the thoughtful comments. Please find below a point-by-point response to these comments, and please find attached our revised manuscript. The changes made in the manuscript in response to the reviewers’ comments are highlighted in red font. We believe that the reviewers’ comments have enabled us to improve our manuscript significantly, and we hope that it is now suitable for publication in Cancers.
Sincerely,
Kazunori Nagasaka, MD, PhD.
Sincerely,
Kazunori Nagasaka
Department of Obstetrics and Gynecology, Teikyo University School of Medicine
2-11-1 Kaga, Itabashi-ku, Tokyo, 173-8605
Tel: +81 3 3964 1211
Fax: +81 3 5375 1274
E-mail: nagasakak-tky@umin.ac.jp
Point-by-point response
Reviewer 1
In figures 2 and 5 ß actin shows that leading was not optimized Figure 4 is not the way to present flow cytometry data, please show "raw FACS data" some curves in figures show statistical significance, some don't HEC1B are mentioned only in methods, otherwise in results are refered only as endometrial cancer cells, which is not very informative
Response: We agree with you, thank you. We added a supplementary figure to show raw FACS data (Supplementary Figure 2). Regarding Figures 2 and 5, unfortunately, we have not demonstrated some experiments using HEC-1B. We have corrected these figure legends accordingly.
Reviewer 2 Report
In the manuscript titled “anti-tumor effect of inhibitions of DNA damage response proteins, ATM and ATR, in endometrial cancer cells”, Takeuchi et al. provided evidence of the effect of combination of ATR inhibitor VE822 and/or ATM inhibitor KU60019 with doxorubicin and cisplatin as well as irradiation in endometrial cancer cells. The experiments are designed well. The data presented support their claims in most cases. Before acceptance, below detailed concerns should be addressed.
Major concerns:
1). In line 90-91, the authors stated that “The IC50 of of VE822 was 1.5 μM and that of KU60019 was 20 μM”. However, Fig. 1 shows IC50 of VE822 is 2.8 μM and that of KU60019 is 22 μM.
2). In Panel C of Fig. 5, 1 μM of KU60019 was added in samples show in lane 3. However, 10 μM of KU60019 was described in text (Line 143 Page 6). It is not clear which concentration is correct from their experiments.
3). Fig 4 shows the effect of adding KU60019 in DXR-induced cell cycle profile change and adding VE822 in CDDP-induced cell cycle profile change. What happens DXR treatment with or without VE822 and CDDP treatment with/without KU60019?
4). In Fig. 8, total H2AX should be used for loading control for gamma-H2AX in the immunoblotting analysis. The immunoblotting experiment in Panel A should be repeated in HEC-6 cells, and the immunostaining analysis of panel B should be repeated in HEC-1B cells.
5). The concentrations of AZD7762 in Fig. 7 is 10, 50, and 100 nM. Why are 30 and 60 nM of AZD7762 chosen in Fig. 8A experiment?
6). The effect of combining ATR inhibitor and Chk1 inhibitor is very interesting. However, the underlying mechanism is not clear from the manuscript.
Author Response
November 25th, 2019
Prof. Dr. Samuel C. Mok
Editor-chief
Cancers
We would like to resubmit our manuscript entitled “Anti-tumor effect of inhibition of DNA damage response proteins, ATM and ATR, in endometrial cancer cells” (Manuscript ID: cancers-647276). We thank the editor and the reviewers for carefully reviewing our manuscript and for the thoughtful comments. Please find below a point-by-point response to these comments, and please find attached our revised manuscript. The changes made in the manuscript in response to the reviewers’ comments are highlighted in red font. We believe that the reviewers’ comments have enabled us to improve our manuscript significantly, and we hope that it is now suitable for publication in Cancers.
Sincerely,
Kazunori Nagasaka, MD, PhD.
Sincerely,
Kazunori Nagasaka
Department of Obstetrics and Gynecology, Teikyo University School of Medicine
2-11-1 Kaga, Itabashi-ku, Tokyo, 173-8605
Tel: +81 3 3964 1211
Fax: +81 3 5375 1274
E-mail: nagasakak-tky@umin.ac.jp
Point-by-point response
Reviewer 2:
In the manuscript titled “anti-tumor effect of inhibitions of DNA damage response proteins, ATM and ATR, in endometrial cancer cells”, Takeuchi et al. provided evidence of the effect of combination of ATR inhibitor VE822 and/or ATM inhibitor KU60019 with doxorubicin and cisplatin as well as irradiation in endometrial cancer cells. The experiments are designed well. The data presented support their claims in most cases. Before acceptance, below detailed concerns should be addressed.
Major concerns:
1). In line 90-91, the authors stated that “The IC50 of VE822 was 1.5 μM and that of KU60019 was 20 μM”. However, Fig. 1 shows IC50 of VE822 is 2.8 μM and that of KU60019 is 22 μM.
Response: Thank you for your positive feedback.
Regarding your concern, we apologize for the mistake. We fitted the dose-response curves as a 4-parameter logistic dose-response model to re-calculate the IC50 value; because we used a different fitting method, the IC50 value has changed. We have updated the results in a new Figure 1.
2). In Panel C of Fig. 5, 1 μM of KU60019 was added in samples show in lane 3. However, 10 μM of KU60019 was described in text (Line 143 Page 6). It is not clear which concentration is correct from their experiments.
Response: Thank you for bringing this mistake to our attention. We used 10 μM of KU60019, not 1 μM of KU60019 as described in the figure. Thus, we have corrected panel C of Figure 5 and the text in the revised version of the manuscript (line 151 page 7).
3). Fig 4 shows the effect of adding KU60019 in DXR-induced cell cycle profile change and adding VE822 in CDDP-induced cell cycle profile change. What happens DXR treatment with or without VE822 and CDDP treatment with/without KU60019?
Response: Thank you very much for question. We performed the clonogenic assay to examine the anti-tumor effect of the combination of conventional chemotherapeutical drugs and these inhibitors. Unfortunately, we have not combined DXR treatment with or without VE822 and CDDP treatment with or without KU60019, as we considered that both DXR and CDDP chemotherapeutic drugs similarly induce DNA damage to cancer cells, and the DDR pathways are inactivated. Therefore, we believe that these combinations (i.e., DXR treatment with or without VE822 and CDDP treatment with or without KU60019) will have the same effect in endometrial cancer cells.
4). In Fig. 8, total H2AX should be used for loading control for gamma-H2AX in the immunoblotting analysis. The immunoblotting experiment in Panel A should be repeated in HEC-6 cells, and the immunostaining analysis of panel B should be repeated in HEC-1B cells.
Response: We agree with your comment. We added new figures to demonstrate the effect of the combination of ATR inhibitor and Chk1 inhibitor. We apologize for the mistake shown in original Figure 8. We have corrected the name of cells accordingly. Unfortunately, we have not performed the immunostaining analysis of HEC-6 cells, because we considered that it may not appropriate to evaluate the effect on HEC-6 cells because of the environment in cell culture.
5). The concentrations of AZD7762 in Fig. 7 is 10, 50, and 100 nM. Why are 30 and 60 nM of AZD7762 chosen in Fig. 8A experiment?
Response: Thank you very much for the comment. In fact, a previous study (ref. 33) shows that this concentration range (30–60 nM) is suitable for the effect. Additionally, 30 and 60 nM are well within the range of the synergistic effect observed at 50 nM of AZD7762 (Fig. 7) and we thus assumed that it would be useful to evaluate the immunoblotting differences, if any, using concentrations immediately above and below 50 nM.
6). The effect of combining ATR inhibitor and Chk1 inhibitor is very interesting. However, the underlying mechanism is not clear from the manuscript.
Response: We agree with this comment. We added further explanation of the effect of ATR and Chk1inhibition in endometrial cancer in the text (line 253-258, page13).
Round 2
Reviewer 1 Report
Changes were well performed
Reviewer 2 Report
none.